



# Alpine Holocene Tree-Ring Dataset: Age-related trends in the stable isotopes of cellulose show species-specific patterns.

Tito Arosio[1,2], Malin M. Ziehmer[1,2,5], Kurt Nicolussi[3], Christian Schlüchter[2,4], Markus Leuenberger[1,2]

[1]Climate and Environmental Physics, Physics Institute, University of Bern, 3012 Bern, Switzerland
[2]Oeschger Centre for Climate Change Research, University of Bern, 3012 Bern, Switzerland
[3]Department of Geography, Universität Innsbruck, 6020 Innsbruck, Austria
[4]Institute of Geological Sciences, University of Bern, 3012 Bern, Switzerland
[5]Swiss Tropical and Public Health Institute, Socinstrasse 57, 4051 Basel, Switzerland

*Correspondence to*: Tito Arosio (tito.arosio@climate.unibe.ch)

**Abstract.** Stable isotopes in tree-ring cellulose are important tools for climatic reconstructions even though their interpretation could be challenging due to non-climate signals, primarily those related to tree ageing. Previous studies on the presence of tree-age related trends during juvenile as well as adult growth phases in δD, $δ^{18}$O and $δ^{13}$C time series yielded variable results that are not coherent among different plant species. We analysed possible trends in the extracted cellulose of tree-rings of 85 larch trees and 119 cembran pine trees, i.e. in samples of one deciduous and one evergreen conifer species collected at the 15   treeline in the Alps covering nearly the whole Holocene. The age trend analyses of all tree-ring variables were conducted on the basis of mean curves established by averaging the cambial-age aligned tree series. For cambial ages over 100 years, our results prove the absence of any age-related effect in the δD, $δ^{18}$O and $δ^{13}$C time series for both the evergreen as well the deciduous conifer species, with the only exception of larch δD. However, for lower cambial ages, we found trends that differ for each isotope and species. I.e., mean $δ^{13}$C values in larch do not vary with ageing and can be used without detrending, 20   whereas those in cembran pine show a juvenile effect and the data should be detrended. Mean $δ^{18}$O values present two distinct ageing phases for both species complicating detrending. Similarly, mean δD values in larch change in the first 50 yr whereas cembran pine between 50-100 yr. Values for these two periods of cambial age for δD and $δ^{18}$O should be used with caution for climatic reconstructions, ideally complemented by additional information regarding mechanisms for these trends.

## 1 Introduction

Stable isotopes in tree-ring cellulose are powerful tools for climatic reconstructions (Kress et al. 2010; Nagavciuc et al. 2019). An advantage of stable isotope time-series based on tree-rings compared to other isotope time series is the defined dating and temporal, e.g. annual, resolution. A challenge for the climatic interpretation of many tree ring parameters is the presence of non-climate signals, primarily those related to tree ageing. Tree-ring width (TRW) and maximum latewood density show ageing effects, which usually have to be removed with detrending/standardization procedures before using them for climatic 30   reconstruction (Helama 2017). A key question for isotope dendroclimatology is whether isotope ratios of the tree cellulose



also show age trends (McCarroll and Loader 2004), still a controversial issue depending on isotope type and plant species. If tree-age related trends are absent the analysis and reconstruction of long-term climatic evolutions based on tree-ring isotope series would loose a source of potential bias.

For δ¹³C, ageing studies were initiated by Freyer et al. (1979) on many tree species, followed by other investigations on , e.g.,
pines, oaks and beeches. All documented an increase for δ¹³C values in the juvenile period (Anderson et al. 2005; Duquesnay et al. 1998; Gagen et al. 2008; Li et al. 2005; McCarroll and Pawellek 2001; Monserud and Marshall 2001; Nagavciuc et al. 2019; Raffalli-Delerce et al. 2004; Treydte et al. 2001). A "long term trend" with continuous increase of δ¹³C values during the entire life of *Pinus sylvestris* was also described (Helama et al. 2015). An early juvenile effect in the first 5 yr was reported for oak (Duffy et al. 2017). Fewer works did not detect any juvenile or long-term trend of δ¹³C values in *Pinus sylvestris* and
larch (Gagen et al. 2007; Kilroy et al. 2016; Young et al. 2011). Altogether, most studies found a juvenile trend (increase), but of variable lengths.

An initial work on δ¹⁸O values found a negative juvenile trend of 300 yr in *Juniperus turkestanica* grown at elevations around 3000 m a.s.l. (Treydte et al. 2006). A similar negative juvenile trend of 300 y was reported also in *Pinus uncinata*, (Esper et al. 2010). A positive trend of 30 yr was found for oak (Labuhn et al. 2014) and positive "long term trends" were reported for
spruce and beeches (Klesse et al. 2018). In contrast, other studies did not find detectable age trends in larch, oak, *Pinus sylvestris*, *Abies alba*, cembran pine (Daux et al. 2011; Duffy et al. 2017; Duffy et al. 2019; Nagavciuc et al. 2019; Saurer et al. 2000; Young et al. 2011). Thus, the studies on δ¹⁸O trends are rather controversial, showing negative, positive or no age-related trends. The mentioned studies analyzed trees from different geographical origins, including some from sites near the treeline where trees often grow in open space. Such growth situations may have an effect on age trends, i.e., under open space
growth there is no neighbourhood competition and therefore no limitation of growth by adjacent trees (Matsushita et al, 2015). The absence of a canopy effect may modify also the isotope composition stored by the trees, as shown for δ¹⁸O (Daux et al. 2011; Esper et al. 2010; Gagen et al. 2008; Nagavciuc et al. 2019; Young et al. 2011). Furthermore, open space growing limits effects described previously such as tree dominance-suppression or location/exposition effects (wet – dry; sunny – shady) (Leuenberger, 2007) that may lead to age trends. The tree height also imposes a hydraulic limitation and possibly reduces
stomatal conductance that may lead to an increase of the cellulose δ¹³C values with increasing age (Brienen et al., 2017).

Changes of δD values in relation to tree ages have been analyzed only in two studies. One found a positive juvenile trend of 20 yr followed by a flat phase in spruce trees (Lipp et al. 1993) and the other identified a positive long-term trend in oak (Mayr et al. 2003).

Helama et al. (2015) suggested that the availability of a suitable database of tree-ring isotopes would allow to detect age trends
and would open possibilities to their elimination in order to improve the recognition of long-term climatic evolution. Five issues were indicated as necessary for such a database: i) that it contains a large sample number, ii) that it has no data from



"pooled" rings, iii) that the samples are well distributed over calendar years with different climate conditions, iv) that the samples come from timberline or treeline sites, where the distance between the trees is large, i.e., limiting the canopy effect, v) that they do not contain "modern" rings that need to be corrected in $\delta^{13}C$ for the anthropogenic rise of CO2 concentration
65 in the atmosphere.

Here we investigated the presence of age trends by utilizing a stable-isotope tree-ring database which was established on the base of the Eastern Alpine Conifer Chronology (Nicolussi et al. 2009). The database consists of i) samples of mainly subfossil wood from 201 trees, ii) the isotope samples are not pooled, iii) the isotope time series with up to multi-centennial length are continuously covering the last ca. 9000 years, iv) it utilizes two different species: deciduous larch and evergreen cembran pine,
70 v) the wood material was collected at different treeline sites and vi) it contains only 17 trees with rings that grew after the industrial revolution. In the present work we aim to verify the presence of age trends in $\delta D$, $\delta^{18}O$ and $\delta^{13}C$ in comparison with those of cellulose content (CC) and TRW. We considered the whole, multi-centennial cambial age range of the trees to identify and quantify the length and extent of the juvenile and the long-term ageing periods.

## 2 Material and methods

75 ### 2.1 Subfossil wood samples and sampling sites in the Alps

Holocene wood sections were available at the Department of Geography of the University of Innsbruck, where the Eastern Alpine Conifer Chronology (EACC) has been established on the base of calendar-dated tree-ring width series (Nicolussi et al. 2009). We have utilized a large number of these subfossil wood samples that cover nearly the whole Holocene (Nicolussi et al. 2009). They belong to the deciduous larch (*Larix decidua Mill.*) and the evergreen cembran pine (*Pinus cembra L.*) and
80 have been collected at treeline sites for paleo-climatic studies. The sampling sites are located in different parts of the European Alps covering a SW-NE transect with an elevation range of 1,930 to 2,400 m (Fig. 1). The wood material was collected at 29 different sites, 3 of them have only larch, 15 only cembran pine and 11 sites contain both species. The characteristics of the 201 trees are listed in Table 1. Only 17 of them contain tree rings formed after the industrial revolution, i.e. after ca. 1850 AD. Samples of 5-year spanning wood have been prepared and analyzed for stable isotope ratios, as described before (Ziehmer et
85 al. 2018; Arosio et al., 2020).

### 2.2 Tree-ring width data and cambial age estimation

The tree-ring width of all the samples were measured with a precision of 0.001 mm as described in Nicolussi et al. (2009). Dated tree samples with relatively wide rings were selected to collect enough material for the isotope measurements. In subfossil specimen the number of rings available for analyses often do not cover the whole tree lifespan due to effects of decay
90 processes and therefore the cambial age of the first measured ring was estimated from ring curvature for tree samples without preserved pith (Nicolussi et al. 2009)





### 2.3 Stable isotope analysis

The procedure of cellulose extraction, determination of the cellulose content (cellulose dry weight / wood dry weight) (Ziehmer et al. 2018) and the triple-isotope analysis were described before (Loader et al. 2015). Briefly, we used conventional Isotope

Ratio Mass Spectrometry (Isoprime 100) coupled to a pyrolysis unit (HEKAtech GmbH, Germany), which is similar to the previously used TC/EA (for technical details see (Leuenberger 2007)). This approach was extended to measurements of non-exchangeable hydrogen of alpha-cellulose using the on-line equilibration method (Filot et al. 2006; Loader et al. 2015). The results are reported in per mil (‰) relative to the Vienna Pee Dee Belemnite (VPDB) for carbon and to Vienna Standard Mean Ocean Water (VSMOW) for hydrogen and oxygen (Coplen 1994). The precision of the measurement is ±3.0‰ for hydrogen,

±0.3‰ for oxygen and ±0.15‰ for carbon (Loader et al. 2015).

### 2.4 Carbon isotope correction

The burning of fossil fuels and land-use changes of the Industrial Period from about 1850 onwards caused a continuous increase of atmospheric carbon dioxide ($CO_2$) depleted in $\delta^{13}C$ (Leuenberger 2007) known as Suess Effect (Suess 1955). This change is reflected in the carbohydrates of the plants, therefore a correction has to be applied to the isotopic series of tree rings. For

all the $\delta^{13}C$ values after the 1000 AD we applied the correction factor described in Leuenberger (2007).

### 2.5 Age trend analysis

Each tree series was aligned on the cambial age and larch and cembran pine samples were analyzed independently. The values of the three isotope ratios were analyzed as raw, normalized and z-scored data. The normalization consisted in subtracting the mean of the time series of a tree from each raw value of this series. For the analysis of the age-related trends, the isotope series

of these three data groups were averaged to mean series for both investigated species under consideration of cambial age of the tree time series. We limited the analysis of these isotope mean series to the cambial age period where the replication number is > = 10 (Klesse et al. 2018; Young et al. 2011). We plotted the normalized data of the isotope values versus the cambial age of the trees. Then we applied a linear interpolation in the different parts of the curves to quantify the trend. The division of the curves is different for each isotope, showcasing their different behaviours. The same procedure was applied to TRW and CC

series, but for TRW we used the raw in place of the normalized data. To verify the presence of trends we applied a linear fit and compared it with those of the isotopes.

### 3 Results

Our aim was to interrogate the stable isotopes of the EACC database for age effects using the well-known age-trends in TRW

as comparison. The plot of the cambial age versus calendar age (Fig. 2A) shows that the database includes isotope time series





covering age ranges from 15 to 610 years, with only few of them starting from the pith. The cambial ages of the trees are rather uniformly distributed over the entire Holocene, thus avoiding a potential bias in the analysis of age-trend. Figure 2A also shows that the time series of the two species, larch (red) and cembran pine (green) are similarly distributed over the Holocene. The sample replication number per cambial age of the trees is shown in figure 2B, with the horizontal line indicating the

threshold≧10, that ranged from 1 to 460yr for cembran pine and from 10 to 480yr for larch. We considered this replication threshold important in this study,

The isotope series of the individual larch trees have a mean length of 273 yr (with a minimum and maximum length of 25 and 550 yr, respectively), the mean of their initial measure is 75 yr and the mean final cambial age is 348 yr. The individual cembran pine time series have a mean length of 257 yr (with a minimum and maximum length of 15 and 610 yr, respectively),

the mean of their initial data point is 67 yr and the mean final cambial age is 324 yr. Table 1 describes the samples we analyzed and shows that they cover the period 8930 b2k to 2010 A.D., with a maximum cambial age of 725 yr.

The means of raw data of each isotope of larch and cembran pine were plotted versus cambial age (Fig 3B, 4B, 5B). The absolute mean values differ probably because of different geographical origin of the trees or species-specific signature (Arosio et al., 2020), therefore we analyzed also the trend with the normalized data. The geographical effect may influence not only

the mean but also the variance of each series, thus altering the age trend. To verify it, we used the z-scored data by dividing the normalized values by the standard deviation for each tree. No consistent difference was found between the normalized and z-score data (supplemental, fig. S1), indicating that the variance of the isotope series was not significantly influenced by geographical factors.

We found that all average isotope series show trend changes only in the first 100 yr of cambial age in agreement with previous

reports (Esper, 2015) except δD of larch. Therefore, we analyzed the average series, before and after 100 yr, separately. The trends of the average series with sample replication above or equal to 10 for all subperiods were studied by linear correlations, as in Young et al. (2011).

### 3.1 δ¹³-Carbon

Means of the standardized $\delta^{13}$C data of all samples from 1 to about 500 cambial yr of both species are shown in figure 3A.

Data points with a replication of $>= 10$ are considered, as shown in the replication plot of figure 2A. The plots of the raw and of the normalized data are shown in figure 3B. The mean raw values of cembran pine (cyan) are more depleted than the ones of larch (red) (Fig. 3B). After normalization the mean values of the two species overlap only partially, since in the juvenile phase of the two species show different trends (fig 3B). Mean values for larch are stable, while the cembran pine documents a strong positive trend in the first 100 yr (0.7 ‰ /100 yr) followed by a stabilization (Fig. 3C).



### 3.2 δ¹⁸-Oxygen

The same approach was used to study the variation of $\delta^{18}O$ of all samples (Fig. 4A) with their mean values in light green. Raw and normalized data are shown in figure 4B. The larch raw data are evidently more $\delta^{18}O$ enriched than the cembran pine ones, and the normalization strongly reduces the difference between the two species and the two time series are almost overlapping (fig 4B, right). In Figure 4C only the normalized averages series of the two species are plotted. Both of them show a peak in the first 100 yr followed by a phase without major age trend. Linear regression was applied to separate an initial phase of 50 yr with increasing values, followed by a sharp decrease until 100 yr and then a stabilization. The initial increasing phase in the larch was less steep than that in cembran pine, and for the rest the patterns are similar.

### 3.3 δ-Deuterium

Means of the standardized δD data of all samples from 1 to about 500 cambial yr of the two species are shown in figure 5A. Means of δD raw values clearly indicate that larch is more depleted than cembran pine (fig. 5B, left) as shown before (Arosio et al. 2020). After normalization the two plots partially overlap (fig 5B, right). Figure 5C shows the means of the two species, the pattern of which are rather different. Larch shows a steep initial decrease in the first 50 yr, after a short steep increase within 10 years the values follows a minor increase through all the time , the cembran pine documents an initial slight decrease in the first 50 yr, followed by a steep positive trend of 50 yr and a flat line from 100 yr on (Fig. 5E).

### 3.4 TRW

The same analysis has also been applied to the non-detrended tree-ring width values, with the difference that in fig 6A the raw data are used. Figure 6A shows all the raw values and the mean value in light green. The plots of the raw and normalized data, expressed in cm, show a similar trend (Fig. 6B). Means with a replication $\geq 10$ (Fig 6B and C) show a maximum at around 30 yr, in agreement with previous reports (Bräker 1981), after which the values of both species steadily decrease in two slope sections until 300 yr, thereafter becoming flat (Bräker 1981). For similarity with the isotope analyses, we applied linear regression to the data, instead of the more common exponential regression.

### 3.5 Cellulose Content

The same analysis has also been applied to data of the cellulose content. Figure 7A shows all raw values and the mean values in light green. The plots of the raw and normalized data, expressed in percent, show a similar trend (Fig. 7B). Means with a replication $\geq 10$ (Fig 7B and C) present a remarkable increase in the first 50 yr in both species, from a cambial age of 51 yr to the end the larch presents a decreasing trend, while the cembran pine shows no trend.



## 4 Discussion

A characteristic of our present work is that the wood samples represent two conifer species, consisting of 201 trees that were
collected at 29 different sites at high elevations in the Alps. They were exposed to different environmental conditions such as,
e.g., elevation, aspect, slope steepness and water availability. Therefore, we normalized all records by subtracting by the mean
of the tree from the raw values (Daux et al. 2011). A different approach was used by Helama et al. (2015) who used the raw
data to analyze samples from only three different sites. Here we present analysis of a much more extended database, in time
and space, which certainly represents the natural variability realistically. However, there are still issues that requires
consideration, in particular the sample replication. Our database does not have a constant sample replication throughout the
cambial age of the trees. It is low at the beginning and increases in the first 50 yr, and decreases sharply after 450 yr. This may
have some effect on the study of the age trends.

As introduced in the results section, we have divided tree ageing in a juvenile period that we deliberately terminated at 100 yr,
and the long-term period that lasted until 450 yr. A major conclusion of this study is that the values of δD, δ$^{18}$O and δ$^{13}$C in
long term period from 100 yr to 450 yr did not change significantly, except δD in larch. This is in agreement with previous
δ$^{13}$C studies on evergreen conifers (Esper et al. 2015; Gagen et al. 2007; Gagen et al. 2008; Klesse et al. 2018; Nagavciuc et
al. 2019; Saurer et al. 2004; Young et al. 2011) and with the δ$^{18}$O previous studies on larch (Daux et al. 2011; Kilroy et al.
2016; Nagavciuc et al. 2019). This implies that no detrending is necessary of tree isotope data for climate analysis with cambial
age in that range, with the exception of δD in larch where a non-significantly trend is present.

More complex are the data of the juvenile period, during which the trend behavior differs among the different isotopes and
species. In the juvenile phase we found evidence for a positive trend of δ$^{13}$C in evergreen cembran pine but not in deciduous
larch. These data are in good agreement with studies on evergreen conifers (*Picea abies, Pinus sylvestris, Pinus uncinata*) that
all found an initial positive trend lasting up to 50 yr (Gagen et al. 2007; Gagen et al. 2008), or 100 yr (Klesse et al. 2018) or
200 yr (Esper et al. 2015). Moreover, two previous studies on larch did not report any evident trend for δ$^{13}$C in the juvenile
period (Daux et al. 2011; Kilroy et al. 2016b). We can conclude that there is a general agreement that deciduous larch and
evergreen conifers behave differently in the juvenile period in respect to δ$^{13}$C of the cellulose. Further work is needed to
understand the reason of this difference.


The behavior of δ$^{18}$O values in the initial period is rather complex, with a maximum around 50 yr and a decrease up to 100 yr
in both species. This is in good agreement with a previous work, that showed, that in *Pinus uncinata* grown at the tree-line
have maximal δ$^{18}$O values around 20-50 yr followed by a negative trend (Esper et al. 2010) and another study on beech and
spruce that found a positive juvenile trend the persisted beyond 50 years of age (Klesse et al. 2018). Altogether, our data show
that δ$^{18}$O of cellulose of larch and cembran pine has ageing trends that are similar to those of other tree species, with up and





down trends in the first 100 yr and an intermediate maximum at around 50 yr. The significance of these trends remain to be further studied. However, considering the time and space of our database covers, this result seems to be widespread and temporally robust.

Our results on δD demonstrate different patterns for larch and cembran pine in the juvenile period, similarly to δ$^{13}$C. The evergreen cembran pine displays an initial flat phase of 25 yr, then an increase of 4‰ till 100 yr. This is in partial agreement with a previous work that showed an increase of δD values in the juvenile period, but this lasted only 20 yr (Lipp et al. 1993). Another work measured δD by nitration of the cellulose to remove non-exchangeable hydrogens (Leavitt 2010), but this is certainly not the reason for this difference, as demonstrated by Filot et al.( 2006). The difference can be attributed to the

different growth environments of the trees, one at an elevation of 330 m near Bad Windsheim (Germany) and the other at a mean altitude of 2100 m, at tree-line sites in the Alps, where tree growth is known to be much slower with prolonged juvenile phase (Körner 2003; Ott 1978). The only other work that analyzed the evolution of δD in cellulose during ageing reported a constant increase of δD values in the first 175 years in oak (Mayr et al. 2003). In our work the deciduous larch showed a different pattern with a strong decrease of δD values in the first 50 yr, followed by a feeble increase smaller than the analytical

precision. We have not found other studies that dealt with δD in larch

The effect of TRW has been studied for long and is well documented (e.g., Helama et al., 2017). After a very short (< 20 yr) period of increasing TRW values, they consistently decrease, yet with different rates. From 20 to 100 yr the decrease is rather steep thereafter changing to moderate rates. At around 300 yr of cambial age they are flattening out in both species.

Understanding the absolute growth rate change is rather complex as discussed for instance in Matsushita et al. (2015). Dependencies of the age-size, growth-size and growth–age relationships are crucial. The fact that our database consists of trees from tree-line sites allows us to state that our derived trends are independent from the so-called crowding effect (influence of neighboring trees). Therefore, it represents mainly the individual variability and the age-size influence on the growth rate. We did not find any dependence of the trends in the different selected time frames within the past 9,000 yr, the behavior we

observed should represent the general dependence of the age-size influence.

The decay of wood does not influence the carbon and oxygen stable isotope values of the cellulose (Nagavciuc et al. 2018), but that it can impact CC, since the cellulose is decomposed faster than lignin. Yet, it has been shown that CC has the potential to be used as a climate proxy (Ziehmer et al., 2018). In the analysis of tree-age related trends we have to consider that the

decay of a trunk is not equal for all parts, and that hardwood in contrast to sapwood presents a decay resistance, which also varies from species to species (Kéérik, 1974). Both investigated species show a similar positive trend in approximately the first 50 yr of cambial age followed by a slight negative trend for larch or no overall trend for cembran pine. This suggests, with reference to our data, that there are probably no (cembran pine) or possibly only minor (larch) influences due to effects of wood decay. This suggests that the CC variation in the first 50 yr is not due to wood decay, but rather a tree-ageing effect.




## 5 Conclusions

The present work confirms the absence of an ageing effect for all three stable isotopes after 100 yr of cambial age in the two conifer species, suggesting that the values older than 100 yr of cambial age can be considered for climate analyses without

detrending. The exception is larch that shows a minor increase of δD mean values, smaller than the analytical precision. Before 100 yr the trends differ for each isotope and species, and only the larch $δ^{13}C$ values can be used without detrending, since they do not vary with ageing. In both species, the $δ^{18}O$ values present two phases, making the detrending rather challenging. It is similar for δD values in larch that change in the first 50 yr, whereas in cembran pine between 50-100 yr. Again detrending is demanding and should ideally be complemented by additional information regarding an explanation of this behavior. Tree ring

cellulose contents show a significant trend for the first 50 years only, in contrast tree ring width curves flattening only after 300 year. Here the application of a regional curve standardization (RCS) is valuable. In summary, for climate reconstructions isotope data older than 100 cambial yr can be use directly, data of the first 100 yr should be used with caution. Therefore, data can be used only after detrending or when compared with data from other age classes covering the same time.

**Acknowledgments.**

We are grateful to Peter Nyfeler for the precious assistance during measurements of the stable isotopes, to Andrea Thurner and Andreas Österreicher for the preparation of the isotope samples from Alpine sites and the civil service collaborators: Lars Herrmann, Giacomo Ruggia, Jonathan Lamprecht, Yannick Rohrer, Rafael Zuber. The project is funded by the Swiss National Science Foundation (SNF 200021L_144255, SNF 200020_172550) as well as by the Austrian Science Fund (FWF, grant I-

1183-N19) and is supported by the Oeschger Center for Climate Change Research, University of Bern, Bern, Switzerland (OCCR).$

*Competing interests.*The authors declare that they have no conflictof interest

## Author contribution statement

TA and MZ performed the stable isotope analyses, TA drafted the first version of the manuscript. KN collected the samples and made the crossdating. ML contributed to the evaluation of the results. ML, KN, CS conceived of the presented idea. All authors provided comments to improve the manuscript.



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




| SITE | SPECIES | N TREES | MEAN LENGTH (YR) | COORDINATES | ASPECT | ELEVATION (m) |
|---|---|---|---|---|---|---|
| AHMO | PICE | 16 | 178 | 47°.03'E/12°08'N | SE | 1995 |
| AHST | LADE | 2 | 125 | 47°05'E/12°11'N | S | 2080 |
| | PICE | 2 | 93 | | | |
| BIH | PICE | 1 | 190 | 46°91'E/10°10'N | N | 2175 |
| EBA | PICE | 11 | 196 | 47°01'E/10°95N | NE | 2115 |
| FPCR | LADE | 4 | 208 | 46°06'E/7°55'N | WSW | 1965 |
| G | PICE | 3 | 215 | 46°85'E/11°01'N | NW | 2060 |
| GDM | PICE | 15 | 197 | 46°88'E/10°71'N | E | 2295 |
| GGUA | PICE | 3 | 160 | 46°85'E/11°N | W | 2175 |
| GLI | PICE | 6 | 164 | 46°81'E/10°7'N | NE | 2147.5 |
| GP | PICE | 5 | 86 | 46°86'E/10°73'N | W | 2167.5 |
| HIB | PICE | 1 | 180 | 46°9' E/12°25'N | E | 2140 |
| KOFL | PICE | 1 | 130 | 46°95'E/12°1'N | S | 2177.5 |
| LFS | PICE | 2 | 148 | 46°81' E/10°7'N | NW | 2335 |
| MAZB | LADE | 6 | 162 | 46°58'E/10°95'N | N | 2125 |
| | PICE | 2 | 198 | | | |
| MAZC | LADE | 4 | 160 | 46°58'E/10°95'N | N | 2120 |
| MAZE | LADE | 5 | 188 | 46°58'E/10°95'N | N | 2125 |
| | PICE | 8 | 124 | | | |
| MAZF | LADE | 2 | 250 | 46°58'E/10°95'N | N | 2105 |
| | PICE | 1 | 155 | | | |
| MIS | LADE | 1 | 125 | 46°9'E/10°61'N | N | 2252.5 |
| | PICE | 1 | 275 | | | |
| MM | LADE | 6 | 247 | 46°03'E/7°916'N | NNE | 1995 |
| | PICE | 7 | 260 | | | |
| MORT | PICE | 3 | 253 | 46°41'E/9°933'N | W | 2045 |
| RT | PICE | 2 | 258 | 46°8'E/10°46'N | SE | 2400 |
| TAH | PICE | 3 | 195 | 46°9'E/11°13'N | S | 2117.5 |
| TSC | LADE | 7 | 206 | 46°4'E/9°88'N | NW | 2162.5 |
| | PICE | 7 | 256 | | | |
| UA | LADE | 17 | 168 | 46°56'E/8°21'N | E | 1950 |
| | PICE | 2 | 205 | | | |
| UAZR | PICE | 4 | 173 | 46°56'E/8°28'N | SSE | 1977 |
| ULFI | LADE | 24 | 243 | 46°46'E/10°83'N | N | 2110 |
| | PICE | 4 | 208 | | | |
| UWBA | LADE | 2 | 200 | 46°46'E/10°81'N | NE | 2330 |
| VRR | LADE | 4 | 219 | 46°43'E/9°85'N | E | 2158.5 |
| | PICE | 5 | 209 | | | |
| ZER | LADE | 1 | 295 | 46°05'E/7°78'N | N | 2315 |
| | PICE | 1 | 110 | | | |

**Table 1. Characteristics of the sampling site and of the trees: 11 sites contain both larch and cembran pine samples, 2 sites contain only larch (LADE) specimens and the remaining sites only cembran pine (PICE)**



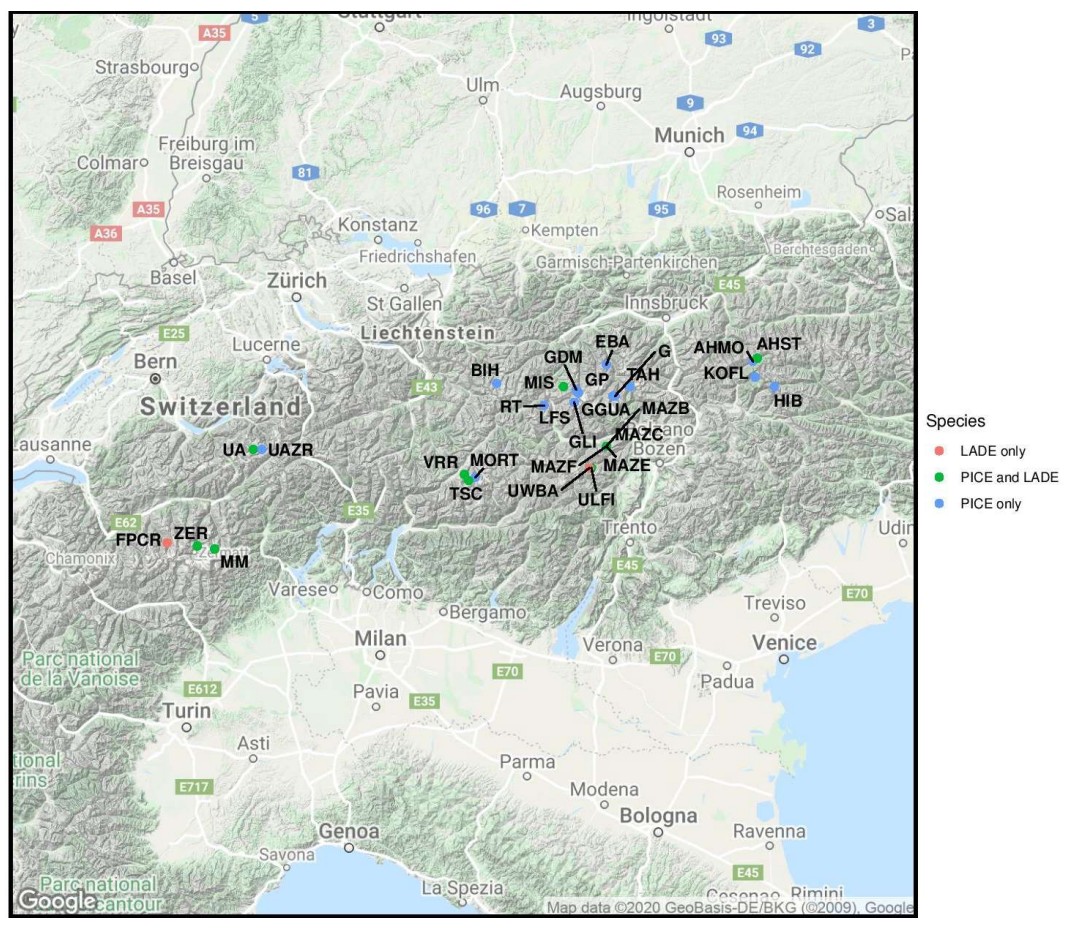

**Figure 1. Map of the location of the 29 sampling sites (© Google Maps 2020): They are situated in the Swiss, the Austrian and the**
**Italian Alps. Information to each site is given in table 1.**



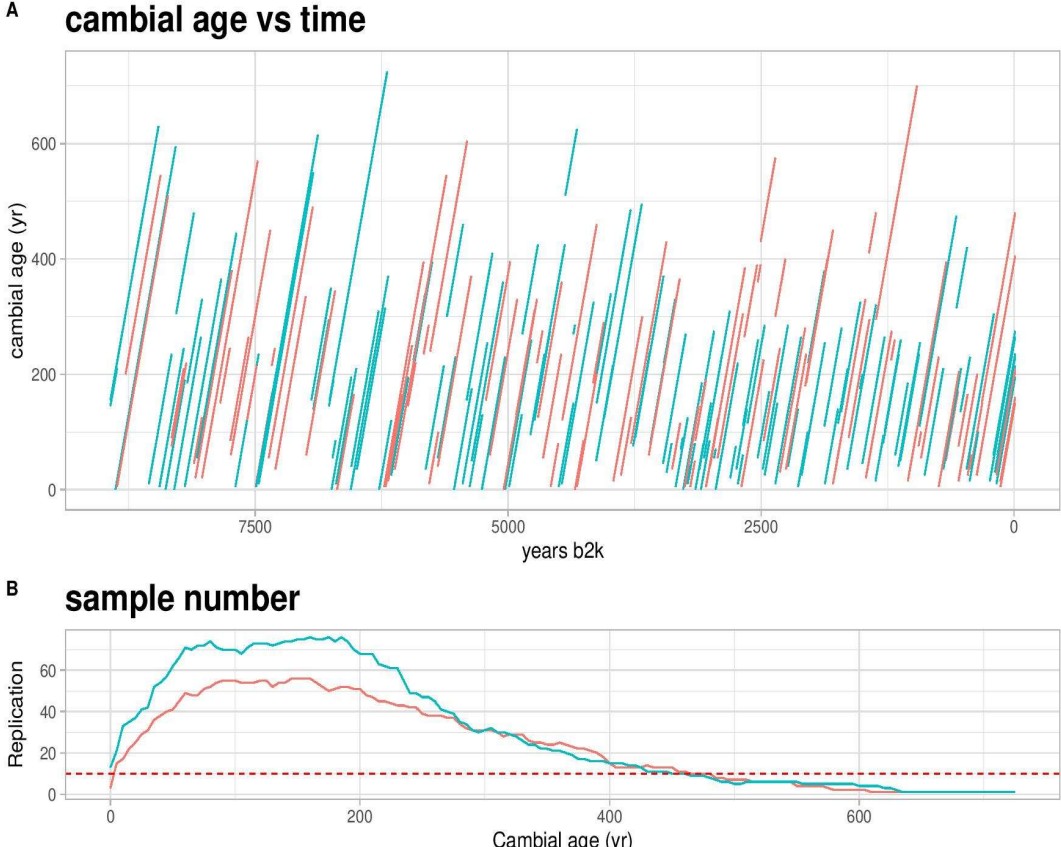

**Figure 2. Cambial age and replication: A, graph of temporal distribution of the all trees and their cambial age. On the X axis the**
**calendar age of the time series is displayed, that goes back to 9,000 yr, on the Y axis the cambial age. Each line represents a tree, in**
**red the larch and in green the cembran pine. B, graph of the replication along the cambial age, the dotted line represents the**
**threshold of 10 that we have considered for the analysis.**


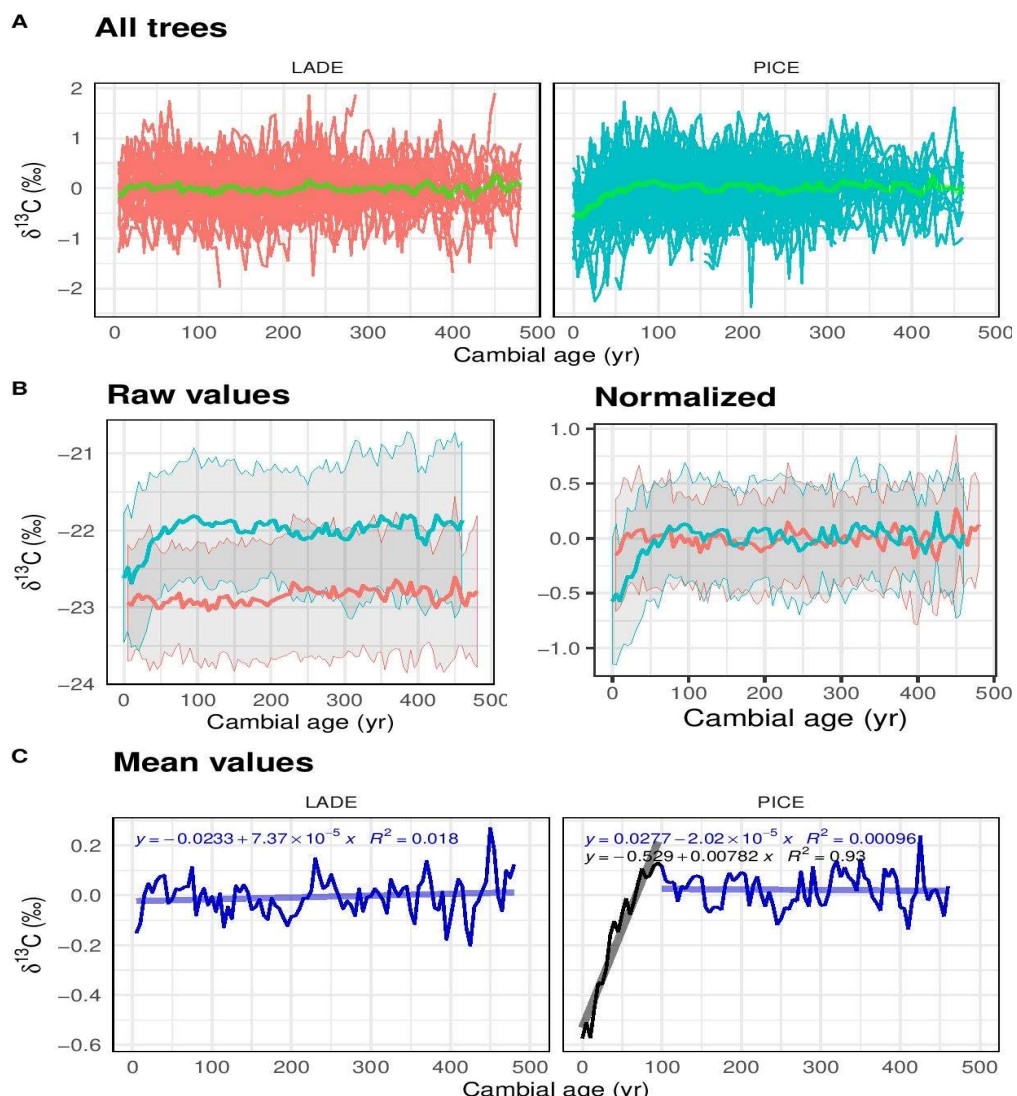

**Figure 3. Analysis of δ13C data: Panel A, normalized δ13C values of all larch (LADE, red) and cembran pine (PICE, green) trees, green line corresponds to the mean. Panel B, raw (left) and normalized (right) mean value with corresponding ± 1 standard deviation (grey area), of larch (red) and cembran pine (cyan). Panel C, plots of the mean values with linear approximations for periods 1-100 yr and >100 yr.**

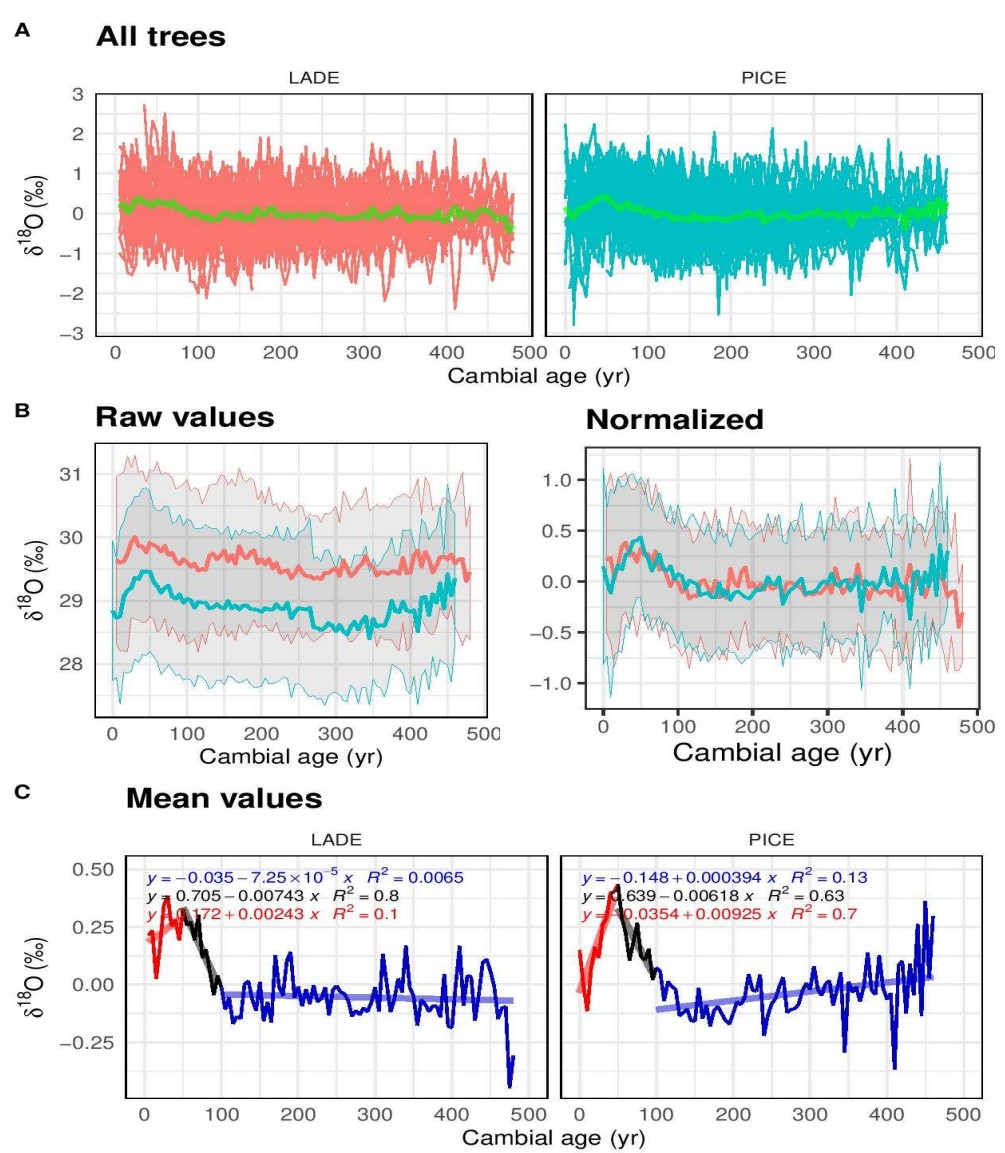


Figure 4. Analysis of δ18O data: Panel A, normalized δ18O values of all larch (LADE, red) and cembran pine (PICE, green) trees, green line corresponds to the mean. Panel B, raw (left) and normalized (right) mean value with corresponding ± 1 standard deviation (grey area), of larch (red) and cembran pine (cyan). Panel C, plots of the mean values with linear approximations for periods 1-50 yr, to 51-100 yr and >100 yr.




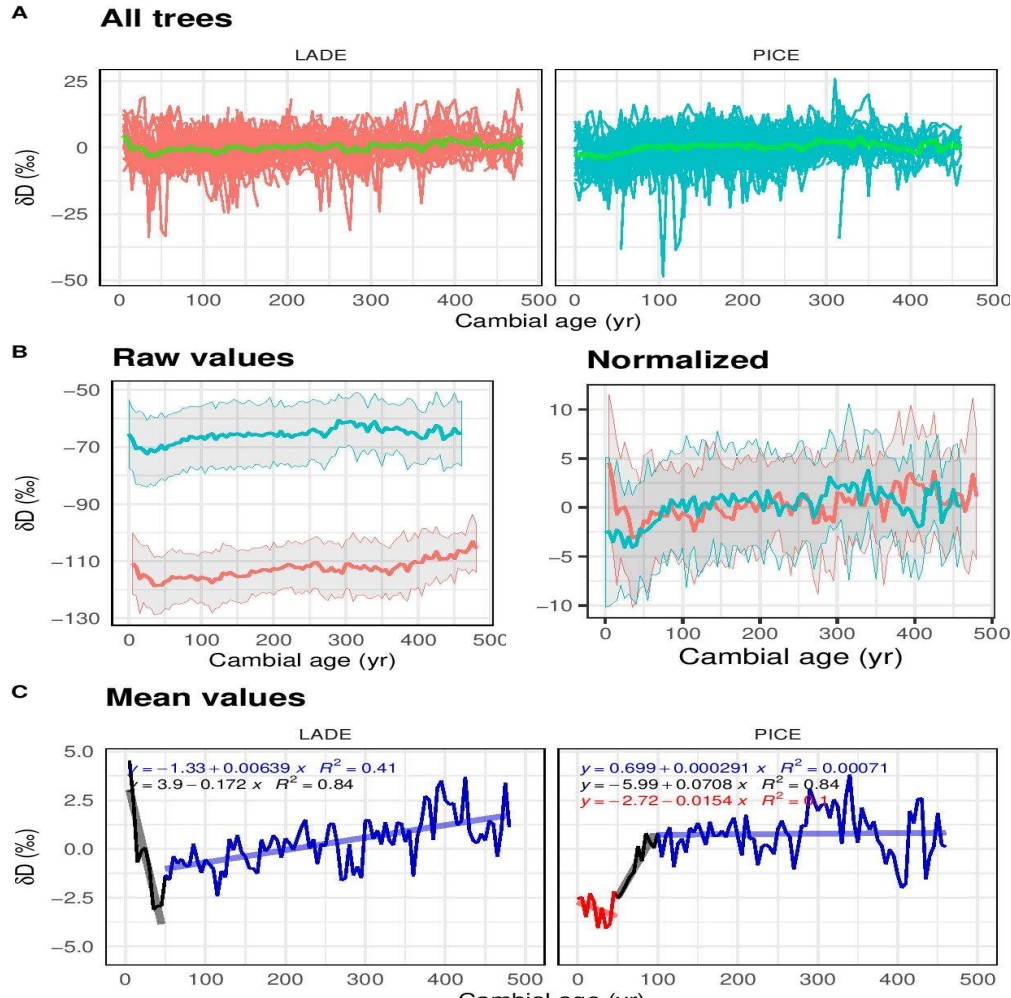

**Figure 5. Analysis of δD data: Panel A, normalized δD values of all larch (LADE, red) and cembran pine (PICE, green) trees, green line corresponds to the mean. Panel B, raw (left) and normalized (right) mean value with corresponding ± 1 standard deviation (grey area), of larch (red) and cembran pine (cyan). Panel C, plots of the mean values with linear approximations for periods 1-50 yr, to 51-100 yr and >100**



EGU Open Access

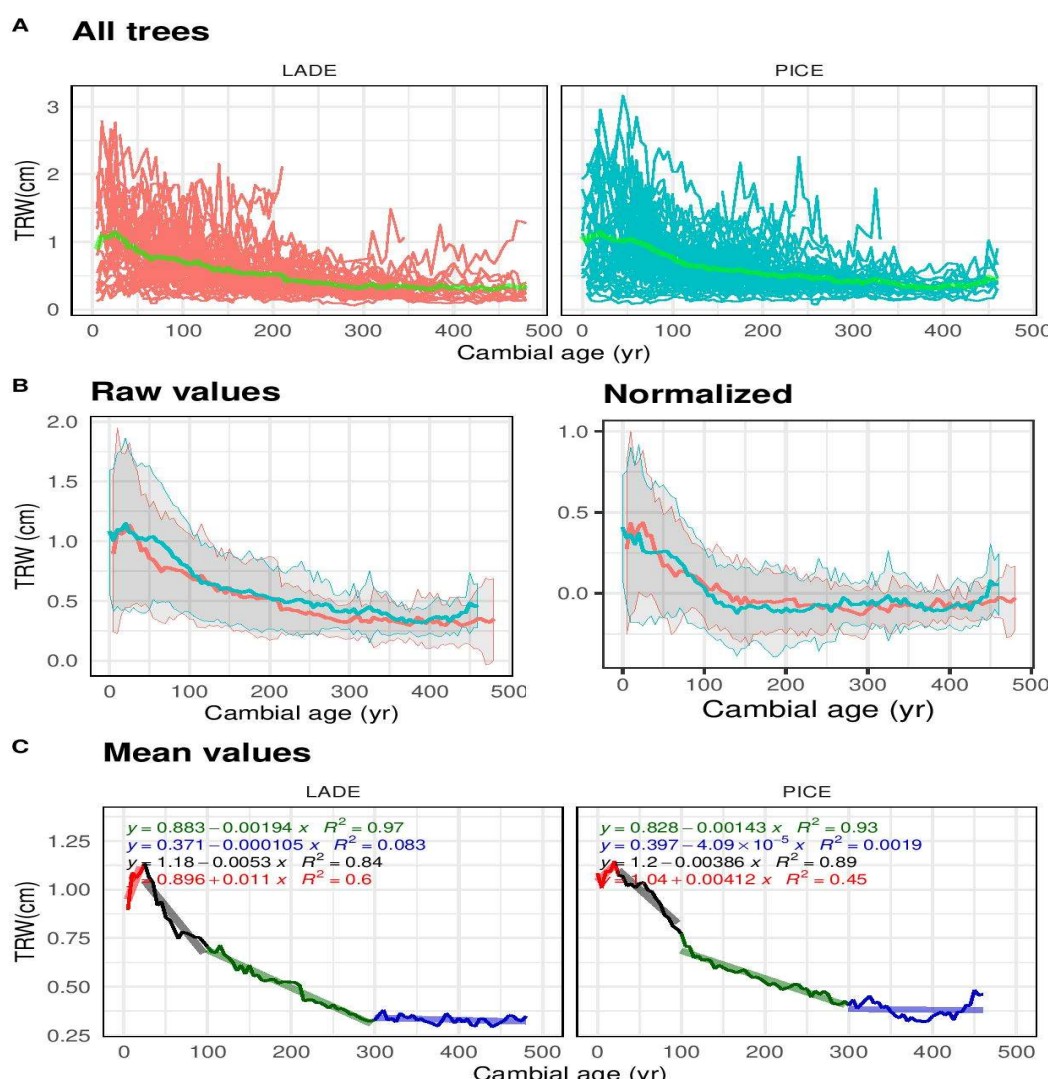

Figure 6. Analysis of Tree Ring Width (TRW) data: Panel A, raw TRW values of all larch (LADE, red) and cembran pine (PICE, green) trees, green line corresponds to the mean. Panel B, raw (left) and normalized (right) mean value with corresponding ± 1 standard deviation (grey area), of larch (red) and cembran pine (cyan). Panel C, plots of the mean values with linear approximations for periods 1-50 yr, to 50-100 yr, to 101-300 yr and >300 yr..



**Figure 7. Analysis of Cellulose Content (CC) data: Panel A, raw value of all larch (LADE, red) and cembran pine (PICE, green) trees, green line corresponds to the mean. Panel B, raw (left) and normalized (right) mean value with corresponding ± 1 standard deviation (grey area), of larch (red) and cembran pine (cyan). Panel C, plots of the mean values with linear approximations for periods 1-50 yr and >50 yr.**