# Peer review of "Alpine Holocene Tree-Ring Dataset: Age-related trends in the stable isotopes of cellulose show species-specific patterns."

_Biogeosciences, 2020_

## Referee Comment (RC1) · Anonymous Referee #1 · 3 Jun 2020

The Papier deals Witz stabile Isotope in trees at treeline Unsinn holocene tree rings. Although ist is a valuable contribution to the Journal, there are some Major flaws. In the introduction the authors should mention why Isotope data Derived from tree rings are powerful Tools for climate Reconstructions There is also evidence that in both Spezies investigated isotope signatures derived from cellulose are comparable to those from bulk wood. See Frontiers in plant science2016,7;799 and agformet 2018,248;251: In fig 1and table 1 please give full namens of all the sites. The results section line 119 - 143 should be moved to methods. In the discussion I miss studies carried out in pine and larch at various treeline Sites in the Swiss and Austrian alps. These data also Show that cembran pine and larch Show similar growt and isotope responses duing the

last decades. This is in contarst to this manuscript and Henne should be discussed.

---

## Referee Comment (RC2) · Anonymous Referee #2 · 23 Jun 2020

The manuscript "Alpine Holocene Tree –Ring Dataset: Age-related trends in the stable isotopes of cellulose show species-specific patterns" by Arosio and co-workers assesses the age-related trends in the stable isotopic compositions of about 200 tree ring samples collected from the Alps. The total period covered by these samples is approximately the Holocene, though individual samples span up to several hundreds of years.

One of the problems in tree ring width analysis is that the ring width often varies with the age of the tree. Thus the non-climatic factors come into play. This problem, to some extent, is encountered in the case of the isotopic records of tree rings as well. Though

the effect of the age on tree ring width is removed by employing statistical techniques, a similar procedure, however, is not so common among the isotope dendroclimatologists. In this study, the authors argue that the age-related trend in isotopic data, especially during the early years of tree growth, is quite prevalent. For example, d18O of tree ring showed inconsistent patterns during the juvenile period of tree growth. So these authors propose a methodology whereby the detrending issue could be avoided. They analyzed a large number of tree ring samples. They suggested that detrending of the tree ring isotopic record is not necessary if the isotopic values are plotted against the cambial age.

But various issues need to be fixed before such a method is applied. For example, which factors are mainly responsible for introducing the so-called age effect in the isotopic values of tree ring cellulose? Why is the result not similar for both oxygen and hydrogen? Since the pathway for both H and O is essentially the same, that is the soil water, then why the d18O and d2H trends sometimes show opposite behavior? Then follows the next question: unless the mechanism is reasonably understood, how can one propose a method to fix the problem? It is known that during environmental stress, such as in a drought season if a particular ring becomes thin, the corresponding d13C would be high. So a long term increasing trend in d13C may be indicative of a prolonged dry condition. But the question may be asked, whether the trend arises as a result of tree ring aging. So the mechanism of causing an age-related trend in an isotopic record must be understood well. In this context, it may be said that the 14C records of tree ring also show age-related trends. But other than the physiological aspect, a physical process, the decay of the radioactive carbon is known to cause such a trend. Hence, by applying a decay correction scheme, the age-related trend on 14C tree ring records can be easily corrected.

In this work, the authors show that isotopic records, especially d18O and d13C, when plotted against the cambial age are not susceptible to age-related trends after a threshold period characterized by juvenile growth. However, the method does not apply to

hydrogen isotopes. Hence it may not offer a comprehensive solution.

As mentioned earlier, long term trends in isotopic values are not considered significantly affecting the paleoclimatic interpretation. Hence they are not typically removed. The authors should give evidence that such practice needs reconsideration so that they can justify their work. Other issues:

The authors have presented all three isotopic records, viz d13C, d18O, and dD, but their inter-relation has not been discussed. d18O and d13C in biological systems (such as corals) often show a strong positive correlation, indicating the presence of kinetic fractionation, and in turn it shows a non-climatic effect. Hence such kind of analysis may be helpful in this context. Similarly, the relation between d18O and dD should also be examined.

Minor issues: Though the manuscript is in general, written well, it contains grammatical errors. (i.e, Line 170-171).

Referencing is not done correctly, pls check Line 308-309.

---

## Author Comment (AC1) · 13 Jul 2020

**Replies to the comments of Referee #1**

The Papier deals Witz stabile Isotope in trees at treeline Unsinn holocene tree rings. Although ist is a valuable contribution to the Journal, there are some Major flaws. In the introduction the authors should mention why Isotope data Derived from tree rings are powerful Tools for climate Reconstructions

Following the reviewer's suggestion we added the following sentence in the introduction: "In environments where the trees are rarely moisture stressed, like at the Alps tree-line, the control of $\delta^{13}$C is the photosynthetic rate, which depends on predominantly on irradiance and temperature. $\delta^{18}$O probably reflects a combination of the direct temperature effect on the isotopic ratio of precipitation and the indirect evaporative enrichment effect. (McCarroll et al. 2003)" We did not mention dD since its paleoclimatic value is still in its infancy and not yet fully understood.

There is also evidence that in both Species investigated isotope signatures derived from cellulose are comparable to those from bulk wood.

It is known that for carbon, oxygen and hydrogen a correlation exists between cellulose and bulk wood (Gori et al, 2013 and Wieser et al, 2016). However, present work uses samples of subfossil wood, in which the wood composition is variably affected by different states of conservation. For this reason, we chose to analyse the cellulose, although it required additional work.

See Frontiers in plant science 2016,7;799 and agformet 2018,248;251

Thank you for the suggestion of the two very interesting papers we missed before. At line 71 we added the following sentence: "This last point is particularly important, because the trees at the Alps tree-line benefit from an enhanced $CO_2$ fertilization and from a recent temperature increase (Wieser et al. 2016) that may produce a long term trend in cellulose isotopes. For example Wieser et al. (2018) show that the increase of temperature is cancelled by increasing atmospheric CO2 concentrations in the environment of the Alps treeline under non-limiting water availability. They state that therefore the instantaneous water use efficiency of photosynthesis did not change considerably."

In fig 1and table 1 please give full names of all the sites

We added the site names in table 1. But the addition of the long and bulky names in Fig. 1 would make it difficult to interpret, so we think it is better to use the codes

The results section line 119- 143 should be moved to methods.

The indicated section presents the properties of the database we interrogated and an initial analysis to understand which of raw, normalized or scaled data are better suited to detect possible trends in the database. So we are convinced that this section belongs to the Result rather than to the Method section.

In the discussion I miss studies carried out in pine and larch at various treeline Sites in the Swiss and Austrian alps. These data also Show that cembran pine and larch Show similar growth and isotope responses during the last decades. This is in contrast to this manuscript and hence should be discussed.

We found only a couple of papers that analysed the age effect in stable isotope in tree rings at mountain treeline, namely Daux et al. (2011) used a larch located at Les Grang, 2050 m.a.s.l. and Esper et al. (2010) used *Pinus uncinata* collected at tree line at about 2300 masl in the Spanish central Pyrenees. This is confirmed by a very recent paper (Büntgen et al, 2020). These papers underlined that the interpretation of the isotopes in the wood of the last decades is particularly complex due to $CO_2$ fertilization. In this study, we used a population of trees that covers the last 9,000 years, which is better suited to understand the non-climatic trends. In addition, these authors do not mention the cambial age of analysed samples. In case their trees were in the adult phase, the data would not be in contrast with our work.

---

## Author Comment (AC2) · 13 Jul 2020

**Replies to the comments of Referee #2**

The manuscript "Alpine Holocene Tree –Ring Dataset: Age-related trends in the stable isotopes of cellulose show species-specific patterns" by Arosio and co-workers assesses the age-related trends in the stable isotopic compositions of about 200 treering samples collected from the Alps. The total period covered by these samples is approximately the Holocene, though individual samples span up to several hundreds of years.

One of the problems in tree ring width analysis is that the ring width often varies with the age of the tree. Thus the non-climatic factors come into play. This problem, to some extent, is encountered in the case of the isotopic records of tree rings as well. Though the effect of the age on tree ring width is removed by employing statistical techniques, a similar procedure, however, is not so common among the isotope dendroclimatologists. In this study, the authors argue that the age-related trend in isotopic data, especially during the early years of tree growth, is quite prevalent. For example, d18O of tree-ring showed inconsistent patterns during the juvenile period of tree growth. So these authors propose a methodology whereby the detrending issue could be avoided. They analyzed a large number of tree ring samples. They suggested that detrending of the tree ring isotopic record is not necessary if the isotopic values are plotted against the cambial age. But various issues need to be fixed before such a method is applied.

We thank the reviewer for the nice summary of our work

For example, which factors are mainly responsible for introducing the so-called age effect in the isotopic values of tree ring cellulose?

Thank you for highlighting an interesting point. Unfortunately, the mechanisms that drive age-effects in the isotopic values of cellulose mainly restricted to the juvenile period are not known. In literature there are many hypotheses mostly related to age-dependent changes in physiology. Our work was aimed at assessing the presence or absence of these effects in the alpine treeline environment regarding three isotopes and two tree species. The differences for $\delta^{13}$C and $\delta$D trends in the juvenile phases of the two species are probably due to different biochemical processes, considering that the trees come from the same environment and that the mean value is the average off all the samples that come from different time epochs and therefore they should not be affected by climate trends. This is presented in the introduction and by fig. 2.

Why is the result not similar for both oxygen and hydrogen? Since the pathway for both H and O is essentially the same, that is the soil water, then why the d18O and d2H trends sometimes show opposite behavior?

Oxygen and Hydrogen originate from the source, i.e. soil water, but they are processed by the trees in a completely different way. $\delta^{18}$O is generally enriched because of evapotranspiration in the leaves, while $\delta$D is generally depleted because of an additional biological fractionation in the photosynthetic processes due to the exchanges with NADH. This is well described by Cormier et al. (2018) (doi: 10.1111/nph.15016) and also in our manuscript ("Larch cellulose is significantly depleted in deuterium isotopes with respect to evergreen conifers" Arosio et al. 2020) that is presently under review in Frontiers in Earth Science.

Then follows the next question: unless the mechanism is reasonably understood, how can one propose a method to fix the problem? It is known that during environmental stress, such as in a drought season if a particular ring becomes thin, the corresponding d13C would be high. So a long term increasing trend in d13C may be indicative of a prolonged dry condition. But the question may be asked, whether the trend arises as a result of tree ring aging. So the mechanism of causing an age-related trend in an isotopic record must be understood well.

The aim of the work was not to propose a method but rather to verify how tree aging in tree line cembran pine and larch affected the tree-ring isotopes, an important issue for paleoclimatic use of the isotopes. It was reported that $\delta^{13}$C is a proxy for conditions of drought (McCarrol and Loader, 2003) but in our case we are analysing a large population of trees covering different climatic periods looking for non-climatic signals. In addition, it is known that at the Alps treeline there is no soil water limitation, due to the frequent rain during the growth season. (Wieser et al, 2016). Also, a major canopy effect on $\delta^{13}$C in the juvenile phase can reasonably be excluded since the two species lived in the same environments.

In this context, it may be said that the 14C records of tree rings also show age-related trends. But other than the physiological aspect, a physical process, the decay of the radioactive carbon is known to cause such a trend. Hence, by applying a decay correction scheme, the age-related trend on 14C tree ring records can be easily corrected.

5  The physical process of decay does not apply to the stable isotopes we analysed, so the aging effect is related only to physiological/biochemical processes, some of which are known to cause major fractionations. They are complex and involve the relationship between autotropism (use of the reserves) and heterotrophism (photosynthesis) which vary during the season and during aging and cannot easily be quantified. Furthermore, the interpretation of the [14]C may also not be so straight forward since it combines
10  the physical decay but also the non-physical, i.e. physiological/biochemical processes. Depending on the weighting of these processes, a simple interpretation of a physical effect only may be misleading.

In this work, the authors show that isotopic records, especially d18O and d13C, when plotted against the cambial age are not susceptible to age-related trends after a threshold period characterized by juvenile growth. However, the method does not apply hydrogen isotopes. Hence it may not offer a comprehensive
15  solution.

The variation of dD after the juvenile period is small indeed, even smaller than the analytical precision of our measurements of 3‰, and the variation was found in larch and not in cembran pine. Aim of the work was to detect the absence or presence of age trends in the two species for the three isotopes, without the ambition to offer a solution of the paleoclimatic use of isotopes.

20  As mentioned earlier, long term trends in isotopic values are not considered significantly affecting the paleoclimatic interpretation. Hence they are not typically removed. The authors should give evidence that such practice needs reconsideration so that they can justify their work.

We do not agree with the statement "isotopic values are not considered significantly affecting the paleoclimatic interpretation". In fact, in the introduction we summarized the literature on the data of stable
25  isotopes for paleoclimatic interpretation, stressing the differences among the isotopes, the conflicting results for $\delta^{18}$O and the limited data for $\delta$D. So there is a lack of agreement about the present or absence of age trends and on how to use the data. For example, recent papers detrended $\delta^{13}$C and $\delta^{18}$O data of tree ring cellulose for a paleo climatic use (Esper et al, 2010 and 2015), while in other papers the values were used undetrended. However, to be clearer, we added a sentence in the introduction quoting examples of papers
30  using undetrended and detrended values of the cellulose stable isotope. "A key question for isotope dendroclimatology is whether isotope ratios of the tree cellulose show age trends or not (McCarroll and Loader, 2004). It is still a controversial issue depending on isotope type and plant species. If tree-age related trends are absent, the analysis and reconstruction of long-term climatic evolutions based on tree-ring isotope series would lose a source of potential bias. Contrary, if they are present, some work suggests to use a
35  detrending procedure (Esper et al. 2010). The same issue concerns the juvenile phase, the values of which can be excluded from paleoclimatic analyses if they show complex age effects; as suggested by (McCarroll and Loader 2004)."

The authors have presented all three isotopic records, viz d13C, d18O, and dD, but their inter-relation has not been discussed. d18O and d13C in biological systems (such as corals) often show a strong positive
40  correlation, indicating the presence of kinetic fractionation, and in turn it shows a non-climatic effect. Hence such kind of analysis may be helpful in this context. Similarly, the relation between d18O and dD should also be examined.

Thank you for the interesting suggestion. Indeed we have been looking at the correlations between the different isotopes and TRW of various cambial age classes, but the preliminary results show that they are
45  complex and their interpretation needs more work, which is outside of the aim of this work. The correlation between $\delta^{18}$O and $\delta^{13}$C in corals is explained by the formation of carbonate, which is very different from the formation of glucose in the photosynthetic and post-photosynthetic processes.

Minor issues: Though the manuscript is in general, written well, it contains grammatical errors. (i.e, Line 170-171).

50  Corrected, thank you

Referencing is not done correctly, pls check Line 308-309.

Corrected, thank you